# Oral Nutrition during and after Critical Illness: SPICES for Quality of Care!

**DOI:** 10.3390/nu12113509

**Published:** 2020-11-14

**Authors:** Marjorie Fadeur, Jean-Charles Preiser, Anne-Marie Verbrugge, Benoit Misset, Anne-Françoise Rousseau

**Affiliations:** 1Department of Diabetes, Nutrition and Metabolic Diseases, University Hospital, University of Liège, Sart-Tilman, 4000 Liège, Belgium; Marjorie.Fadeur@chuliege.be; 2Multidisciplinary Nutrition Team, University Hospital, University of Liège, Sart-Tilman, 4000 Liège, Belgium; am.verbrugge@chuliege.be; 3Erasme University Hospital, Medical Direction, Université Libre de Bruxelles, 1070 Brussels, Belgium; Jean-Charles.Preiser@erasme.ulb.ac.be; 4Department of Intensive Care and Burn Center, University Hospital, University of Liège, Sart-Tilman, 4000 Liège, Belgium; benoit.misset@chuliege.be

**Keywords:** critical illness, nutrition rehabilitation, food intake, quality of care, oral nutrition

## Abstract

Malnutrition is associated to poor outcomes in critically ill patients. Oral nutrition is the route of feeding in less than half of the patients during the intensive care unit (ICU) stay and in the majority of ICU survivors. There are growing data indicating that insufficient and/or inadequate intakes in macronutrients and micronutrients are prevalent within these populations. The present narrative review focuses on barriers to food intakes and considers the different points that should be addressed in order to optimize oral intakes, both during and after ICU stay. They are gathered in the SPICES concept, which should help ICU teams improve the quality of nutrition care following 5 themes: swallowing disorders screening and management, patient global status overview, involvement of dieticians and nutritionists, clinical evaluation of nutritional intakes and outcomes, and finally, supplementation in macro-or micronutrients.

## 1. Introduction

Insufficient feeding (i.e., intakes lower than losses or expenditure) is frequent in critically ill patients, especially for long stayers in intensive care unit (ICU). Moreover, many patients are admitted with a previous altered nutritional and metabolic status that can be further aggravated by prolonged underfeeding [1]. Thus, critically ill patients should be screened for malnutrition as early as admission, at least using general clinical assessment [2]. In the critical care population, low nutritional intakes are associated with poor outcomes, such as a prolonged ICU or hospital length-of-stay (LOS), a higher incidence of complications (such as infections) and, ultimately, an increased mortality [3,4,5,6]. On the contrary, overfeeding can also increase the risk of complications and should thus be avoided [7]. Some authors introduced the concept of “nutritrauma” to raise awareness of the harmful effects of inappropriate nutrition support [8].

Long stayers often experience metabolic, neuroendocrine, and nutritional derangements that are initially triggered by the primary insult and perpetuated by the unresolved failure and persistent inflammation. This situation has been called chronic critical illness (CCI) [9]. These patients carry a high rate of mortality. On another hand, survivors of a critical illness can be affected by the post-intensive care syndrome (PICS), a newly recognized clinical entity characterized by various deficits in physical, cognitive, and/or psychological functioning [10]. An underpinning aspect of these two syndromes is muscle wasting and weakness, at least partly due to high muscle protein catabolism [11]; mitochondrial dysfunction and myopathy [12]; prolonged sedation and lack of physical activity [11]; and impaired muscle regeneration [13].

The available literature on nutrition during critical illness raised awareness of the benefits of medical nutrition. Nowadays, nutrition is considered as a major supportive therapy in critically ill patients. Although there is little evidence for the role of increased nutrition delivery in reducing muscle wasting and improving the recovery of critically ill survivors, there is a physiological rationale to consider nutrition in the post-ICU period as equally important as during the ICU stay [14]. 

## 2. Oral Nutrition during and after Critical Illness

About 40% of the critically ill patients are able to eat during their ICU stay [15]. Trajectories of orally fed patients in ICU are heterogeneous: oral diet can be exclusive during the entire ICU stay, can be transitioned from a parenteral or enteral nutrition (such as after tracheal extubation), or can be provided simultaneously to enteral nutrition in awake patients.

According to the few studies published so far, critically ill patients who were fed orally had very low intakes in term of both energy and proteins, compared to predicted requirements and recommendations [5,16,17,18,19,20], regardless of the underlying cause. In the different studies, energy intakes varied from 30 to 50% of the daily requirement or prescription. Protein intakes were even lower and did not reach 40% of the daily requirement or prescription. To the best of our knowledge, some studies focusing, at least partly, on oral nutrition in critically ill patients are currently active or recruiting worldwide (Table 1). This is encouraging. However, oral nutrition could still be considered as a comfort care, rather than a key component of critical care. When patients look less severely ill, oral nutrition is less closely monitored, and critical-illness-related malnutrition can occur or worsen. The insufficient consideration is reinforced by the fact that we have no idea of the impact on recovery of a mismatch between intakes and needs.

Oral nutrition is not only a question of dose to be ingested. To fuel cells, nutrients have to be absorbed. It has been shown that critically ill patients may exhibit a decreased or delayed gastrointestinal functional absorptive capacity [21,22]. This may limit the beneficial effects of oral nutrition, despite providing adequate quantities.

In the latest update of their recommendations on nutrition practices during critical care, scientific societies highlighted the specific nutritional needs, both quantitatively and qualitatively, of the critically ill patients [2,23]. However, these recommendations are mainly focused on enteral or parenteral nutritional support, during the ICU setting. Due to the paucity of data on oral nutrition during or after an ICU stay, recommendations on oral diet are mostly experts’ opinion. This lack of evidence is partly related to the inconsistency of ICU discharge timing and step-down units availability among hospital and research groups, precluding comparisons between different studies.

Currently, there are limited research, and consequently recommendations, on nutrition in the post-ICU period. Oral nutrition provided alone is the most common mode of nutrients provision during this period. During the post-ICU hospital setting, it has been observed in two recent studies that energy and protein intakes were less than estimated or measured requirements [24,25]. On the other hand, we are partly blinded on what happens to ICU survivors who are discharged at home without any follow-up in terms of nutritional adequacy and related outcomes.

## 3. Strategies to Optimize Oral Nutrition during and after Critical Illness

Multiple factors contribute to undernutrition in critically ill patients or survivors. Barriers to consuming adequate nutrition are mainly elated to swallowing disorders, reduced appetite, and food access or food services. Moreover, prescription or delivery of macro- and micronutrients may be inadequate, quantitatively or qualitatively. The “SPICES” concept (Figure 1) is a mnemonic acronym conveying the factors and subsequent strategies that can help reaching adequate food intakes in orally fed patient throughout the entire journey from critical illness to recovery. The concept is composed of the five following themes, described below: swallowing disorders screening and management, patient global status overview, involvement of dieticians and nutritionists, clinical evaluation of nutritional intakes and outcomes, and finally, supplementation in macro- or micronutrients.
S: Swallowing disorders screening and management

Swallowing dysfunction is induced by damage to the central nervous system, tracheal intubation, and presence of a nasogastric tube or a tracheostomy [26]. The duration of mechanical ventilation seems to be a prominent risk factor [27]. Reported incidences are highly variable, depending on the patient selection, the timing, and the method of assessment. The frequency of long-term mild-to-severe dysphagia in ICU survivors of severe sepsis reaches the range of stroke survivors [28]. However, this important concern still suffers from limited awareness and screening [29]. Typical clinical signs evoking dysphagia are listed in Table 2. Further assessment of dysphagia is based on screening questionnaires, clinical observation, and bedside non-instrumental evaluation. A number of tools are available: they generally involve the swallowing of a volume, defined in terms of quantity and viscosity. Even if they were initially proposed for non-ICU patients, some of the tools, such as the volume-viscosity swallow test (V-VST), are safely suitable in the acute care setting [30]. Final diagnosis is obtained using instrumental tests, such as oropharyngo-esophageal scintigraphy or flexible endoscopic evaluation [26,29]. The latter is considered as the gold standard technique, but its use should be strictly limited in patients with coronavirus disease 2019 (COVID-19) [31].

Dietary texture modification is a major therapeutic pillar for dysphagia. The food textures (and drink thickness) are described using the International Dysphagia Diet Standardization Initiative (IDDSI) Framework terminology [32,33]. The type of texture is prescribed according to the results of the swallowing assessment. However, these modifications can lead to patient dissatisfaction and reduced intakes. The variety of food that can be modified is quite limited, and meals are then uniform. Moreover, switching food to (semi-)liquid textures lowers the nutrients density. Texture modification should be associated to active deglutition rehabilitation, guided by speech-language therapist, and including postural changes, compensatory maneuvers (i.e., special swallowing techniques), and therapeutic exercises. Pharyngeal electrical stimulation is an invasive treatment approach, at present mostly used in stroke-related dysphagia [29].
P: Patient global status overview

Many factors related to patient status can interfere with oral nutrition. Not all can be changed, managed, or treated. At best, they should all be considered in order to adjust the nutritional strategy.

Appetite is commonly reduced, from early during critical illness, to at least 3 months after discharge [34,35]. A relationship between nutritional intakes and levels of gut released peptides has been described. In critically ill patients, peptide YY, a hormone that inhibits appetite, has been shown to be high, while ghrelin, a peptide that stimulates appetite, was low [36]. Beside appetite changes, taste can be altered by medications or viral infections, such as COVID-19; satiety can be modified; and gastric emptying can be impaired. Moreover, patients can further suffer from nausea, pain, anxiety, delirium, or sleep disturbances. Muscle weakness and reduced physical capacity have secondary effects on chewing fatigue and ability to self-feed. Finally, food intake is driven by individual, social, or environmental determinants. 

Eating preferences and habits or religious food restrictions can reduce the list of palatable foods or the qualitative composition of meals. Social isolation and absence of commensality are well-known factors contributing to poor eating behaviors, as predominantly demonstrated in the elderly population [37]. Such conditions are easily reproduced in ICU, especially in units without open visitation policy, where patients eat alone. The eating environment (light, noise, odor, seating comfort) impacts food consumption [38] but is not always adequate in hospitals. Unfortunately, impact of all these factors on nutritional intakes in critically ill patients have not been widely studied.
I: Involvement of dieticians and nutritionists

In hospital, another category of barriers to food intake are organizational factors [39]. Food delivery can be inappropriate in terms of timing in patients who suffer from sleep disturbances. Portion size may be inadequate in view of reduced appetite. Basic appearance, inappropriate temperature, microwave heating, or bland taste can make the dishes unappetizing. Moreover, some meals can be missed due to mandatory fasting for invasive procedures or surgeries. Traditionally, access to food is limited between meals, and ordering systems are poorly flexible [40].

Dieticians/nutritionists have a key role in optimizing oral nutrition during ICU and hospital stay [19]. They assist critical care and hospital teams by improving the feeding process in terms of food access and delivery. They also help to formulate the oral nutritional intervention, aiming to individualize the nutritional support. A clear nutritional plan should be established, based on the patient’s food desires and nutritional requirements. This plan should thus be further communicated to ward teams, to general practitioners, and finally, to patients and families, who may lack specialized knowledge about the complex requirements of a recovering ICU survivor. A closed follow-up by the dietetic team may help ICU survivors to reach adequate nutritional intakes during their recovery.
CE: Clinical Evaluation of nutritional intakes and outcomes

Metabolic response and, subsequently, nutritional strategies vary according to the critical illness trajectory. Even if still vague from a clinical point of view, three phases may be described [2,41]: after injury or insult, the acute phase is composed of an early and a late period, before progressing to late phase (Figure 2). Macronutrient intakes during the acute phase are progressively increased to reach at least 70% of energy expenditure (EE) (or 20–25 kcal/kg/day) and maximum 1.3 g/kg/day proteins between day 3 to day 7 at the latest [2,42]. These European recommendations are valid for either medical nutrition or oral nutrition. During the late phase, corresponding to either CCI and/or convalescence and rehabilitation, higher protein intakes and energy targets may be assumed. As an example, some authors [43] suggest, during the post-ICU phase at hospital, to increase energy intakes to 125% of the predicted requirements or to daily provide 30 kcal/kg and to increase protein intakes to 1.5–2 g/kg/day. After post-hospital discharge, energy intakes could be increased to 150% of the predicted requirements or to provide 35 kcal/kg energy per day. During that recovery period, protein intakes could even be higher than 2 g/kg/day. These targets make sense based on the available literature but still need to be validated in post-ICU studies.

At best, energy targets should be determined using repeated indirect calorimetry [2] that measures inspired and expired gas exchanges to calculate EE. Even if its benefits on outcomes are not established [44], critically ill patients are good candidates for indirect calorimetry [45]. It is now recognized that predictive equations are not accurate enough and that metabolic status of critically ill patients or survivors are highly dynamic during their ICU path, with high inter-individual variability [45]. New generation of calorimeter allow easily and accurate measurement of EE in spontaneously breathing patients [46,47]. Unfortunately, indirect calorimetry using canopy is not always feasible (i.e., in case of oxygen supply) or widely available. Simpler appreciations are derived from predictive equations. It is noteworthy that these predictive equations are associated with significant inaccuracy, leading to inadequate evaluation of the energy needs [48]. The least inexact equation is the Penn State equation for ICU patients [49]. The simplest estimation of energy requirements are weight-based equations, as described in Figure 2.

Nutritional interventions should be monitored, especially as needs are very dynamic throughout the critical illness journey. First, it is important to monitor what the patient really receives and eats and assure that nutrition is provided as planned and prescribed. Yet, as stated above, nutrition delivery or intakes are often inferior to prescription. Daily and cumulated energy and protein balances should be calculated (net balance = target − delivery). In hospital, this can be performed advantageously using computerized management systems [50,51]. Through such tools, the nutrition delivery has been shown to be closer to the targets [52]. For survivors’ follow-up, telemedicine is another option: text messaging or smart-phone applications have become widely used interfaces between patients and providers. Mobile-based dietary assessment tools are emerging and promising, but the accuracy of their database still needs to be improved [53]. Secondly, nutrition interventions should be evaluated for effectiveness. How the patient responds to the delivered nutrients is another key assessment. Nutrition-related outcomes include biomarkers. Serum albumin and prealbumin are traditional biomarkers thought to reflect nutritional status. However, due to their relationship with systemic inflammation response, they are not sensitive markers of energy and protein intake adequacy [54]. It might be useful to observe the evolution of body composition, muscle architecture, muscle strength, and physical function from ICU to full recovery [55]. The frequency of repetitions has not been determined.
S: Supplementation in macro- and/or micronutrients

Nutritional targets may be found enormous by patients, compared to the amounts they are able to ingest. In order to achieve macronutrients targets in ICU patients who receive oral nutrition and in ICU survivors, food enrichment or oral nutrition supplements (ONS) will often need to be considered [24,43,56]. This is particularly the case for patients with dysphagia who need liquidized (IDDSI 3), pureed (IDDSI 4), or minced and moist (IDDSI 5) foods. Food such as cream, yogurt, fruit purée, mashed potatoes, milk, coffee, or soup can be enriched by adding maltodextrin or protein powders. These powders are neutral in taste. Maltodextrin powder contains around 380 kcal/100 g. Protein powder contains around 90 g protein/100 g. ONS can be either liquid (drinks) or, less frequently, semi-solid (creams). Drinkable ONS are milk or fruit juice derived. A number of commercial products are available, varying in term of flavor, fiber composition, and volume. They contain macronutrients and micronutrients at varying levels of concentrations. Calories generally range from 150 to 400 kcal/portion, and protein concentration ranges from 8 to 20 g/portion. Generally, ONS are prescribed one or twice daily, respecting patient’s flavor choices. Information regarding ONS intake should be provided to patients: they should be advised to take ONS between meals, aiming to avoid spoiling appetite. 

Especially during ICU or hospital stay, patients are not able to ingest sufficient enriched food or ONS to reach the macronutrients targets. In these patients, a supplemental enteral feeding should be considered. In particular, after extubation when there is a high incidence of swallowing disorders and weakness, the naso-gastric tube should probably not be hastily removed, at least until oral intakes ensure adequate energy and protein provision.

Deficits in micronutrients are suspected to be frequent in ICU survivors, as a result of severe illness itself, drug interactions, increased consumption, or extensive losses of biologic fluids. Iatrogenic losses may be related to sedation administration [57] or to insufficient repletion during continuous renal replacement therapy (CRRT) [58]. Magnitude of such deficits is not well-documented in the current literature [59]. Moreover, micronutrient status is difficult to approach: routine lab assessment is not common and is expensive, and the plasma concentrations are influenced by systemic inflammatory response [60,61]. In the latter case, concentrations should be interpreted with caution as low values do not necessarily reflect deficiency, and normal values do not necessarily indicate satisfactory bioavailability or bioefficiency.

Micronutrients are vital as intermediaries in metabolism and have potential roles in wound healing, cellular immunity, and oxidative stress. If possible, they should come from a well-balanced diet. However, micronutrient requirements may not be covered by oral diet alone, due to suboptimal eating patterns or temporary increased requirements [62]. Unlike burn patients for whom some recommendations are published [63], appropriate dose, route, and length of supplementation are unknown or controversial in critical ill patients. At least, the provision of dietary references should be guaranteed [64]. Especially in trauma patients, in burn patients, or in the case of wound healing disorders, tailored nutritional measures or supplementation in micronutrients could be beneficial [65] (Table 3). Supplementation or repletion in some micronutrients can be administered orally, separately, or as commercial preparations of multivitamins and minerals. Unfortunately, the oral route can be associated with reduced bioavailability or competition between trace elements (such as zinc and copper). Other at-risk populations are known to have altered micronutrient status, such as patients with severe acute kidney injury (regardless of treatment with CRRT) [66], alcoholic patients, patients on proton pump inhibitors or metformin, and patients who benefited from a bariatric surgery (Table 3) [62,67,68,69]. In these situations, the effect of micronutrients supplementation still needs to be explored.

Notwithstanding, the lack of dedicated recommendations, there are some rationales for some micronutrients to be continued after ICU or hospital discharge. According to recent data, iron deficiency, diagnosed using hepcidin, is very frequent at ICU discharge and has been associated to poor one-year physical recovery [70]. Indeed, iron deficiency causes fatigue and muscle weakness, independently of anemia [71]. Magnesium and selenium correlate to muscle mass and muscle performances [72]. 

Vitamin D insufficiency or deficiency is very common in critically ill patients and has been associated with adverse outcomes [73]. Many of these patients enter the ICU with a pre-existing deficient status. However, hypovitaminosis D can be induced by the ICU stay and the critical illness themselves. The cause is multifactorial: lack of exposure to sunlight, inadequate intakes, malabsorption, and altered metabolism [74]. Among the pleiotropic effects of vitamin D, its beneficial effect on immunity or muscles is undebated [75,76]. A normal vitamin D level requires regular vitamin D supplementation in many individuals, typically in a daily dose of 600 to 2000 IU [77]. There is now evidence that higher replacement dosage may be required in critically ill patients, compared to other populations [78]. The benefit of supplementation may vary according to the vitamin D status. In the VITdAL-ICU study, high doses of cholecalciferol reduced 28-day mortality only in patients with very low levels of 25(OH)-D [79]. Moreover, preliminary reports suggest a beneficial effect of vitamin D3 supplementation in increasing quadriceps strength in adults with vitamin D deficiency [80], as well as in the post-burn period [81,82].

## 4. Perspectives in Nutrition Therapy and Metabolic Support

Adjuvant therapies to nutritional support are increasingly discussed, aiming to optimize nutritional support or modulate metabolic status in ICU patients or survivors. Orally fed patients during ICU stay or after discharge could theoretically benefit from such strategies. However, only a few data are available up to now to guide their prescription, particularly the optimal timing of introduction. Depending on future evidence, metabolic support could become one of the features of “supplementation”, as an upgrade of the last “S” of the SPICES concept.

Carnitine is an amino acid derivate, which places a crucial role in fatty acid transport and β-oxidation, and at least, in mitochondrial energetic functioning [83]. Carnitine deficiency should be suspected in chronic critically ill patients and in patients who were dependent on CRRT or parenteral nutrition. Relevant signs include unexplained hypertriglyceridemia, fatty liver, or myopathy [54]. Carnitine can be measured in plasma: low level is an additional argument for supplementation, at least orally (0.5–1 g/day) [84].

Ghrelin agonists are a therapeutic perspective to help overcome the reduced appetite experienced by critically ill patients and survivors. According to preliminary studies in animals or some groups of patients, benefits could be multiple: improvement of food intake, but also the additional effect on the upregulation of the growth hormone axis, or the modulation of the inflammatory response [85]. Unfortunately, in a recent study in critically patients (although enterally fed), the ghrelin agonist ulimorelin failed to be superior to metoclopramide in term of gastric emptying [86]. More data are needed before ghrelin modulation could be implemented in clinical practice.

Nutrition rehabilitation should probably be viewed as a part of a multimodal strategy aiming to enhance recovery. Conceptually, such a strategy should at least combine adequate nutrition and active physical exercises [87]. Evidence is still weak [88,89]. In a recent study that should be interpreted with caution due to protein target in the control group, authors observed that a high protein regimen provided better muscle mass maintenance only with associated active rehabilitation [90]. Anabolic agents, anticatabolic agents, and specific nutrients could be additional and innovative approaches, aiming to modulate muscle wasting [87]. It is thought that the five approaches could be synergistic. However, in the absence of strong evidence in critically ill patients, such a combination is not yet systematically recommended. Probably, it should rather be prescribed on an individualized basis. 

The rationales for prescribing anabolic agents, and particularly testosterone and analogues, are the sustained catabolism and the anabolic resistance observed in ICU survivors [13,91]. This ongoing metabolic status is associated with testosterone deficiency [92], further increased by opioid use [93]. Anabolic agents have been more widely studied in severe burn patients: oxandrolone, an androgen analogue, safely induces a gain in lean body mass during the rehabilitative phase [94]. Administration of recombinant human growth hormone (rhGH) is not recommended in critically ill adults. Unlike in the general intensive care population [95], no adverse impact on mortality was observed in burn patients. However, in burn adults, rhGH effects were not better than oxandrolone [96], while disclosing adverse hyperglycemia. At present, patients who will benefit the most from anabolic treatments are not clearly identified, as well as the optimal timing of administration or even dosage.

The majority of the published data about benefits of anticatabolic agents arises again from the severe burn population. Treatment with non-selective beta-blockers (i.e., propranolol) reverses muscle protein catabolism in severe burn children [97] and does not impede exercise-induced increases in muscle strength [98]. In an old study including head-injured patients, propranolol also reduced the observed resting hypermetabolism [99]. Unfortunately, since those princeps studies, there has been no significant advances with this treatment in non-burn critically ill patients.

The amino acid leucine is known to promote muscle protein synthesis [100]. β-Hydroxy-β-methylbutyrate (HMB), a metabolite of leucine, improves muscle mass and muscle strength in a variety of clinical groups [101]. Type (leucine or HMB) and dose of amino-acid supplementation is still unknown, as well as the best phase of critical illness in which to provide this.

## 5. Conclusions

Malnutrition is a significant concern for the critically ill patients who are fed orally. ICU survivors remain vulnerable, at least for months following discharge. In the absence of dedicated recommendations about macro- and micronutrients, careful nutritional management of oral nutrition during and after an ICU stay should appeal to common sense. Any problem limiting food intakes should be addressed. A closed monitoring of daily intakes, their adequacy with nutritional requirements, and nutritional-related outcomes should be set up early and prolonged after discharge. Intervention of a multidisciplinary team including a dietician is crucial, helping to raise awareness among ICU caregivers about difficulties in feeding critically ill patients orally. There is an urgent need for studies clarifying the condition of orally fed patients during and after ICU stay.

## Figures and Tables

**Figure 1 nutrients-12-03509-f001:**
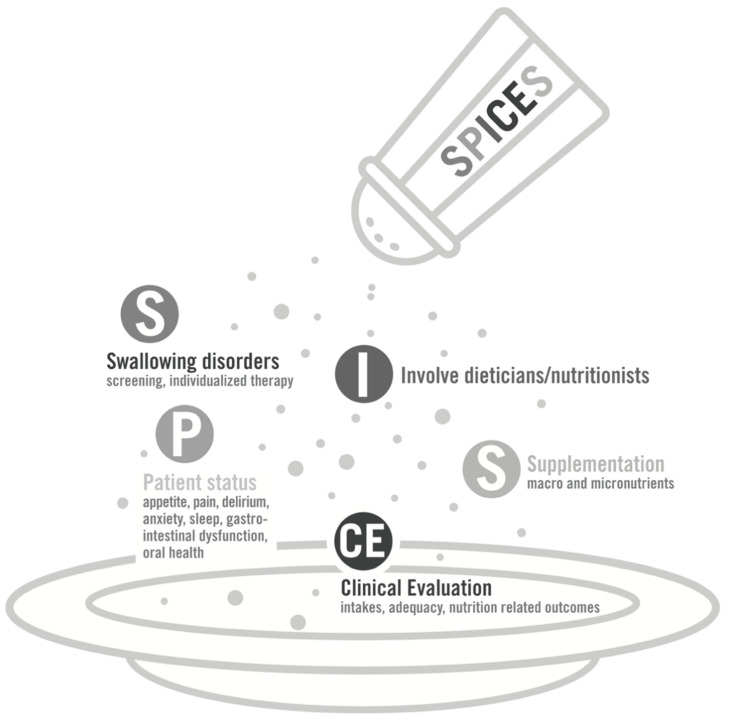
SPICES concept, detailing the different points to be addressed in a critically ill patient or an intensive care unit (ICU) survivor, aiming to optimize oral feeding.

**Figure 2 nutrients-12-03509-f002:**
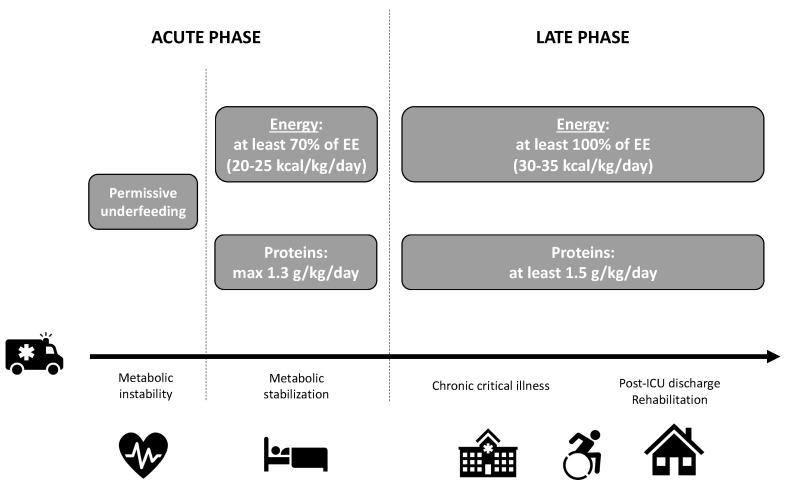
Calories and proteins provision by oral route, according to the successive phases of the critical care pathway (adapted from [23,39,41]). Abbreviations: EE, energy expenditure; max, maximum.

**Table 1 nutrients-12-03509-t001:** Planned or ongoing studies including, at least partly, adult critically ill patients or survivors on oral nutrition (sources: ClinicalTrials.gov (https://www.clinicaltrials.gov); EU Clinical Trial Register (https://www.clinicaltrialsregister.eu); Japan Primary Registries Network including JMACCT (http://www.jmacct.med.or.jp/en/ctr/ctr_list_p8.html), UMIN (https://www.umin.ac.jp/ctr/), and JapicCTI (http://www.japic.or.jp/); Australian New Zealand Clinical Trails Registry (https://anzctr.org.au); ISRCTN registry (https://www.isrctn.com)—28 October 2020 and 9 November 2020).

Register Identifier	Design	Region	Inclusion Criteria	Primary Outcome	Secondary Outcomes	Intervention	Comparator	Planned Sample Size
NCT 04549961	Observ.	Austria	Admitted to ICU on nutritionDay.	60 days hospital mortality. Length of ICU stay. Route of nutrition. Planned and delivered amount of nutrition in kcal from all routes.	Number of ICU beds. Human resources.	NA	NA	3500
NCT 04274322	Observ.	China	Anticipated length of ICU stay >48 h. Diagnosed with 2019 coronavirus disease (COVID-19). Food intake difficulties.	28-day all-cause mortality.	From admission to 28 days after discharge: All-cause infection. Rate of complications. Length of ICU stay. Duration of mechanical ventilation.	NA	NA	117
NCT 02920086	PRT open label	Canada	1/ For patients: >60 years of age OR 55 years to 59 years old with one or more comorbidities. Projected duration of ICU dependency of >72 h from time of final assessment. 2/ For family member: Present and expected to visit regularly. The nominated or legally appointed substitute decision-maker. Able to communicate in English.	Nutritional adequacy during the ICU stay. Consumption of ONS. Caloric intakes on hospital wards. Hand grip strength. Use of shared decision-making. Change in decisional conflict. Family satisfaction with decision-making. Overall family satisfaction with ICU.	Not provided.	1/Nutrition Education Program involving families. 2/Decision support program involving families.	Usual care	150
ACTRN12620001025921p	Observ.	Australia and New Zealand	Adults admitted in ICU for >48 h.	Energy prescription and intakes. Feeding intolerances. Delivery of dietetic services.	Protein prescription and intakes. Descriptions in nutrition practice variability across Australia and New Zealand. Hospital length-of-stay. Infectious complications.	NA	NA	500
ACTRN12620000602921	Observ.	Australia	Patients included in Short Period Incidence Study of Severe Acute Respiratory Infection (SPRINT-SARI): adults with a suspected or proven acute respiratory infection requiring new inpatient admission with onset within past 14 days.	Nutrition service delivery. Nutrition provision. Proportion of patients diagnosed with malnutrition using any of the validated malnutrition screening and assessment tools.	Data about nutritional management during prone position.	NA	NA	200
UMIN 000040290	Interv. Single arm, open and not randomized	Japan	Adults admitted in ICU, with an expected hospital stay ≥10 days.	Femoral muscle volume change from day 1 through 10	MRC score, FSS-ICU, EQ-5D at ICU discharge. Barthel Index at hospital discharge. Target nutrition achievement rate N-titin level of days 1, 3, 5, and 7.	Rehabilitation (including electrical stimulation of lower limbs) and nutrition administration.	Historical control.	50
UMIN 000042057	Observ.	Japan	Adults admitted in ICU for >3 days.	Protein/non protein calories ratio, from ICU admission to the day before hospital discharge.	Muscle mass and Barthel Index the day before hospital discharge.	NA	NA	180

Abbreviations: NA, not applicable; Observ., observational; Interv., interventional; ONS, oral nutritional supplement; PRT, prospective randomized trial; ICU, intensive care unit.

**Table 2 nutrients-12-03509-t002:** Clinical signs of dysphagia.

Difficult or Painful Chewing or Swallowing
Regurgitation of undigested food.
Difficulty of controlling solids or liquids in the mouth.
Drooling.
Coughing during or after swallowing.
Feeling of obstruction.
Frequent throat clearing.
Recurrent bronchitis or pulmonary infections.

recurrent items in non-instrumental assessment tools.

**Table 3 nutrients-12-03509-t003:** Frequent clinical situations at risk of micronutrient deficiency and the corresponding suggested supplementation during ICU stay or in ICU survivors (adapted from [62,69]).

Clinical Situation	Micronutrient Supplementation
Suboptimal eating patterns. Energy intake < 1500 kcal/day.	Multivitamins/multiminerals, vitamin D, calcium, vitamin B12, magnesium.
Prolonged ICU and/or hospital stay.	Vitamin D, calcium.
Prolonged wound healing, polytrauma.	Multivitamins/multiminerals.
PPI (long term treatment).	Vitamin B12, magnesium, calcium.
Severe acute kidney injury, CRRT.	Multivitamins/multiminerals.
Persistent kidney disease.	Vitamin D, vitamin K.
Post-bariatric surgery status.	Multivitamins/multiminerals, vitamin D, calcium, vitamin B12, iron.
Alcohol abuse.	Vitamins B, fat-soluble vitamins, zinc.
Liver disease (i.e., fatty liver).	Zinc, selenium, vitamins A, D, K, and B12.

Abbreviations: CRRT, continuous renal replacement therapy; ICU, intensive care unit; PPI, proton pump inhibitor.

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
