# Peer review of "Oral Nutrition during and after Critical Illness: SPICES for Quality of Care!"

_nutrients, 2020, doi:10.3390/nu12113509_

Round 1
Reviewer 1 Report
This is the review article which discuss the number of problems related to oral nutrition during and after critical care and how to resolve it in future. I completely agree with the authors' opinions and this is one of the important issue in critical care nutrition which would interest the medical staffs in critical care. However, I think the whole manuscript should be revised according to the main concept of this article. Major and minor concerns were written as follows.
1. The manuscript consist below component.
2. Oral nutrition during and after critical illness
3. Barriers to food intakes
4. Strategies to optimize oral nutrition
5. Perspectives in nutrition therapy and metabolic support
6. Don’t forget micronutrients!
I have no opposing opinion about these components. However, considering the aimed theme "Oral nutrition during and after critical illness: SPICES for quality of care!", the author should discuss more about 2,3,4, especially for 4. In the section 2, they should review and discuss details about all the published study about oral nutrition and it's intake amounts during and after critical care, which would be the keys of this theme. They should try to consider the facts which can be extracted from this review and connect them to 4.
2. Moreover, if the authors want to propose the word of SPICE and it's approach, they should discuss the details more about each components of S, P, I, CE. I recommend that each components S, P, I, CE should be divided into separate sections and described/discussed how to achieve them in more detail with adequate references. Meanwhile, I think the nutrition support approaches such as enhancement of appetite or metabolism would be important for securement of oral intake after critical care in the future, as the authors discussed in the other section. Which is such a strategy in SPICE approach?
3. Chronic critical illness CCI is not the concept focusing on nutrition status. The author should discuss carefully about the correlation between CCI/PIICS and nutrition.
4. The introduced future studies in Table.1 are too less than in truth. Why did they examine only in Clinicaltrials.gov? They should try to find out in the other trial registration sites. I believe there would be more ongoing trials in the world.
5. It is too old to say the final version of ESPEN and ASPEN guidelines as "recently". Recent recommendation may be changed from these final version with the recent articles.
6. Reference numbers are wrong (in all? some deviations exist).
7. I am afraid that the deficiency of micronutrients in each clinical condition are not adequate in table.3. The author should add the references to each clinical conditions and revise all with adequate information.
Reviewer 2 Report
Thank the authors for this article, which highlights the relevance of nutritional therapy at all times of the disease and in its recovery. Indeed, the continuity of the optimization of nutrition in an individualized way is fundamental.
Although they will allow me to disagree on different points to improve:
- GLIM criteria in critical patients has not shown its usefulness and just as the NUTRIC score (only validity scale) has many limitations so that you can recommend it.
- In this same journal the concept NUTRITRAUMA (deleterious effects of applying inadequate nutrition) was published, perhaps starting to use this term makes the objective of your article more evident. (Yébenes JC, Campins L, Martínez de Lagran I et al. Working Group on Nutrition and Metabolism of the Spanish Society of Critical Care. Nutritrauma: A Key Concept for Minimising the Harmful Effects of the Administration of Medical Nutrition Therapy. Nutrients. 2019 Aug 1;11(8):1775. doi: 10.3390/nu11081775. PMID: 31374909; PMCID: PMC6723989).
- When you talk about the clinical signs of dysphagia, you forget earlier data of suspicion that we can obtain when applying the MECV-V(Clinical Evaluation Method Volume-Viscosity) in the critical patient. Its safety criteria are very interesting for the critical patient and its application at the bedside, even more so.
- I think you should include this review article on dysphagia. Frajkova Z, Tedla M, Tedlova E, Suchankova M, Geneid A. Postintubation Dysphagia During COVID-19 Outbreak-Contemporary Review. Dysphagia. 2020 Aug;35(4):549-557. doi: 10.1007/s00455-020-10139-6. Epub 2020 May 28. PMID: 32468193; PMCID: PMC7255443.
- In my modest opinion, your article is based on the ESPEN guidelines and repeats the same thing, perhaps it would be of greater interest to talk about its applicability, because you already know that the best is sometimes the enemy of the good:
- Indirect calorimetry is not yet available even in most UCIS and there is still time to have enough experience to apply it properly, it still has clear limitations. Hopefully its usefulness will be real as soon as possible.
- Few patients with PICS have the capacity to ingest 150% of their caloric intake and 2 g of protein orally in the recovery phase. We have no scientific evidence to recommend it. In my opinion, what they consider an expert opinion seems to me a personal opinion. In case you're right, ESPEN's guidelines probably fall short on input? It seems prudent to wait for more evidence to promote this clinical practice. Quantity or quality??
- They have evaluated the possibility of recommending as another measure to maintain the nasogastric tube at ICU discharge to promote complementary enteral nutrition, (for example at night and I am aware that it is less physiological). As you well say, the patient has to recover from a great aggression. Some clinicians interpret the removal of the gastric tube with greater healing, which is a mistake. Without a doubt, it should be individualized, but as some authors say, the gastric tube should not be removed until intake is greater than 60% (american dietitians). Patients do not eat the diet or oral supplements. The monitoring of this phase is undoubtedly essential to make it more dynamic, because the days go by and the patient becomes malnourished. I would dare to say that he loses even what he has not lost in the ICU because the possibilities of monitoring and optimization are less.
- About macro and micronutrients little can be said more, insist on monitoring.
- By last, you have an error in the enumeration of the bibliography (from number 21), please check it .
Encourage the authors to carry out a multicenter study collecting patients upon discharge from the ICU, perhaps we continue to fail in the most basic thing.
Round 2
Reviewer 1 Report
The manuscript has been revised and improved much better responding to my suggestion. Minor concerns are following.
- I am afraid yet that table.1 review was insufficient. It would be impossible to detect the critical care study in which oral intake would be evaluated or not definitely. What is the Japanese registry network? UMIN? On the UMIN, I-GREEN study, in which critical care nutrition is evaluate with evaluation of oral intakes, is registered. According to this concern, the author should discuss not that there were few studies, but that there were a few studies, as positive meaning appreciating to their works.
- I agree with the response "We fully agree with the reviewer regarding the perspectives in nutrition support and metabolic modulation. However, to date, these strategies suffer from insufficient evidence. This precludes any recommendations about their introduction in routine practice. We thus chose to be careful and to describe these perspectives in separate paragraph.". However, I believe that metabolic modulation or pharmaco-nutrition approaches would possibly become an important approach in the future with the future evidence. I want to suggest to discuss the possibility to add it in SPICE, for example to add last s "SPICES"; s is the support of metabolic modulation/pharmaco-nutrition in future, if the related evidence would be structed.
